

**Comparison of key absorption and optical properties between pure**
**and transported anthropogenic dust over East and Central Asia**
Jianrong Bi[1], Jianping Huang[1*], Brent Holben[2]
[1]Key Laboratory for Semi-Arid Climate Change of the Ministry of Education, College of
Atmospheric Sciences, Lanzhou University, Lanzhou, 730000, China
[2]NASA Goddard Space Flight Center, Greenbelt, Maryland, USA
*Submitted to: ACP Special Issue*

-----------------------------------------------

*Correspondence to:* Jianping Huang (hjp@lzu.edu.cn)





**Abstract.** Asian dust particulate is one of the primary aerosol constituents in the Earth-atmosphere system that exerts profound influences on environmental quality, human health, marine biogeochemical cycle and Earth's climate. To date, the absorptive capacity of dust aerosol generated from Asian desert region is still an open question. In this article, we compile columnar key absorption and optical properties of mineral dust over East and Central Asia areas by utilizing the multi-year quality assured datasets observed at 13 sites of the Aerosol Robotic Network (AERONET). We identify two types of Asian dust according to threshold criteria from previously published literature. (I) The particles with high aerosol optical depth at 440 nm ($AOD_{440} \geq 0.4$) and low Ångström wavelength exponent at 440-870 nm ($\alpha < 0.2$) are defined as Pure Dust (PDU) that decrease disturbance of other non-dust aerosols and keep high accuracy of pure Asian dust. (II) The particles with $AOD_{440} \geq 0.4$ and $0.2 < \alpha < 0.6$ are designated as Transported Anthropogenic Dust (TDU), which are mainly dominated by dust aerosol and might mix with other anthropogenic aerosol types. Our results reveal that the primary components of high AOD days are predominant by dust over East and Central Asia regions even if their variations rely on different sources, distance from the source, emission mechanisms, and meteorological characteristics. The overall mean and standard deviation of single-scattering albedo, asymmetry factor, real part and imaginary part of complex refractive index at 550 nm for Asian PDU are $0.935 \pm 0.014$, $0.742 \pm 0.008$, $1.526 \pm 0.029$, $0.00226 \pm 0.00056$, respectively, while corresponding values are $0.921 \pm 0.021$, $0.723 \pm 0.009$, $1.521 \pm 0.025$, and $0.00364 \pm 0.0014$ for Asian TDU. Aerosol shortwave direct radiative effects at the top of the atmosphere (TOA), at the surface (SFC), and in the atmospheric layer (ATM) for Asian PDU ($\alpha < 0.2$) and TDU ($0.2 < \alpha < 0.6$) computed in this study, are a factor of 2 smaller than the results of OPAC Mineral accumulated (Mineral acc.) and transported (Mineral tran.) modes. Therefore, we are convinced that our results hold promise of updating and improving accuracies of Asian dust characteristics in present-day remote sensing applications and regional or global climate models.



## 1. Introduction


Airborne dust particle (also called mineral dust) is recognized as one of the most
important aerosol species in the tropospheric atmosphere, which accounts for about
30% of the total aerosol loading and extinction aerosol optical depth on a global scale
(Perlwitz et al., 2001; Kinne et al., 2006; Chin et al., 2009; Huang et al., 2014). High
concentrations of dust aerosols hanging over desert source regions and invasive
downstream areas would seriously exacerbate air quality, degrade visibility, affect
transportation safety, and do adverse effects on public health during the prevalent
seasons of dust storms (Chan et al., 2008; Morman and Plumlee, 2013; Wang et al.,
2016). When mineral dusts are deposited onto the Earth's surface, they play a key role
in biogeochemical cycles of terrestrial ecosystem or ocean (Okin et al., 2004; Jickells
et al., 2005; Shao et al., 2011), as well as alter snow and ice albedo (Aoki et al., 2006;
Huang et al., 2011; Wang et al., 2014). Last but not least, dust particles can modulate
the Earth's energy budget and drive the climate change directly by scattering and
absorption of solar/terrestrial radiation (Charlson et al., 1992; Wang et al., 2010b;
Huang et al., 2014), and indirectly by acting as effective cloud condensation nuclei or
ice nuclei, influencing the cloud microphysics and precipitation processes
(Ramanathan et al., 2001; Rosenfeld et al., 2001; DeMott et al., 2003; Huang et al.,
2005, 2006, 2010a; Wang et al., 2010c; Creamean et al., 2013). Numerous studies
(Sokolik and Toon, 1999; Lafon et al., 2004, 2006) have confirmed that dust particle
is one kind of light absorbing substances, and its mass absorption efficiencies at 325
nm (0.06~0.12 $m^2$/g) are about 6 times larger than at 660 nm (0.01~0.02 $m^2$/g), owing
to the greater absorbing potential of iron oxides at short wavelengths (Alfaro et al.,
2004). However, the way of iron oxides mixed with quartz or clay is complicated and
strongly impacts the resulting absorption (Claquin et al., 1998, 1999; Sokolik and
Toon, 1999). And these mineralogical studies indicate that a lack of consideration of
these mixing mechanisms is a significant limitation of the previous dust absorption
computations. Although the absorptive ability of dust is two orders of magnitude
lower than for black carbon (Yang et al., 2009), the atmospheric mass loading of the





former is the same magnitude larger than that of the latter, leading to the total
absorption in solar spectrum comparable to black carbon. Chin et al. (2009) evaluated
that dust may account for about 53% of global averaged aerosol absorption optical
depth at 550 nm, which undoubtedly changes the aforementioned
dust-cloud-precipitation interaction and exerts a significant effect on hydrological
cycle of the Earth-atmosphere system.

East and Central Asia territories are the major source regions of dust aerosols on

Earth, which produce a large amount of dust particles every year that become
entrained into the upper atmosphere by cold fronts (Zhang et al., 1997; Huang et al.,
2009, 2010a, 2014). They can travel over thousands of kilometers, even across the
Pacific Ocean and reach the western coast of North America about one week with the
prevailing westerly wind (Husar et al., 2001; Uno et al., 2009, 2011), and then modify
the climate and environment over extensive area of Asia-Pacific rim. Thus far, there
have been a great deal of fruitful field campaigns for exploring Asian dust (e.g.,
U.S.S.R.-U.S., ACE-Asia, ADEC, PACDEX, EAST-AIRC), however, most focus on
intensive observation period (Golitsyn and Gillette, 1993; Huebert et al., 2003;
Nakajima et al., 2003; Mikami et al., 2006; Huang et al., 2008a; Li et al., 2011) and
lack of long-term and quantitative knowledge of dust optical, microphysical
characteristics (especially absorption properties) and chemical compositions over
these regions. Hence, the absorptive capacity of Asian dust aerosol is still an
outstanding issue. The variations of dust optical features in model calculations are
closely related to the uncertainties in particle size distribution and prescribing a value
for complex refractive index. Whereas the key parameters of Asian dust aerosols in
present-day climate models are still prescribed to the predetermined properties of
Saharan mineral dust.

Wang et al. (2004) inferred the refractive index of pure minerals at Qira in

Taklimakan Desert during April 12-14, 2002 via combination of theory calculation
and composition analysis of aerosol samples, and showed that the value of imaginary
part is 0.00411 at 500 nm, which is consistent with the Central Asian dust of
0.004±0.001 (Tadzhikistan Desert; Sokolik and Golitsyn, 1993). Uchiyama et al. (2005)



determined the single-scattering albedo (SSA) of Aeolian dust from sky radiometer
and in situ measurements, and concluded that unpolluted Aeolian dust (source from
Taklimakan Desert) has low absorption (with $SSA_{500}$ of 0.93~0.97). Kim et al. (2004)
analyzed multiyear sky radiation measurements over East Asian sites of
Skyradiometer Network (Nakajima et al., 1996; Takamura et al., 2004) and showed
the $SSA_{500}$ of dust particles are around 0.9 in arid Dunhuang of northwest China and
Mandalgovi Gobi desert in Mongolia. Bi et al. (2014) also reported the similar $SSA_{550}$
(0.91~0.97) of dust aerosol at Dunhuang during spring of 2012. Xu et al. (2004)
gained $SSA_{530}$ of 0.95±0.05 in Yulin, China, from a Radiance Research nephelometer
and a Particle Soot Absorption Photometer (PSAP) and suggested that both desert dust
and local pollution sources contributed to the aerosol loading in Yulin during April
2001. Whereas Ge et al. (2010) examined dust aerosol optical properties at Zhangye
(a semiarid area of northwest China) from multifilter rotating shadowband radiometer
(MFRSR) during spring of 2008 and found that although there are low aerosol optical
depth values ($AOD_{670}$ ranging from 0.07~0.25), dust particles have strong absorption
(with $SSA_{500}$ of 0.75±0.02) due to mixing with local anthropogenic pollutants. This
result is close to the New Delhi over India (0.74~0.84 for $SSA_{500}$; Pandithurai et al.,
2008). Lafon et al. (2006) revealed that due to containing of less calcite and higher
fraction of iron oxide-clay aggregates, mineral dusts in Niger (Banizoumbou, 13°31′N,
2°38′E) have much lower SSA in the visible wavelengths than that of Chinese (Ulan
Buh, 39°26′N, 105°40′E) and Tunisian (Maouna, 33°01′N, 10°40′E) desert locations.
Therefore, complete clarification of the climate-relevant impacts of Asian dust
aerosols requires extensive and long-term measurements of the optical, microphysical
and chemical properties, along with their spatial and temporal distributions.

In this paper, we investigate optical characteristics of Asian dust from multi-year

AErosol RObotic NETwork (AERONET) measurements at 13 sites in and around arid
or semi-arid regions of East and Central Asian desert sources. The key quantities
include single-scattering albedo (SSA), asymmetry factor (ASY), real part (Re) and
imaginary part (Ri) of complex refractive index, volume size distribution (dV/dlnr),
which are needed for climate simulating and remote sensing applications. We mainly



compare the vital absorption and optical properties between pure and transported
anthropogenic dust over East and Central Asia. This article is arranged as follows.
Section 2 introduces the site description and measurement. The identification method
and detailed Asian dust optical features are described in Section 3. Discussion of
spectral absorption behaviors of different dust aerosol types are given in Section 4 and
followed by the Summary in Section 5.

## 2. Site Description and Measurement

### 2.1. Site Description

In this article, we select 13 AERONET sites located in arid or semi-arid Asian
regions (see Fig. 1), which are recognized as the primarily active centre of dust storms.
These drylands are very sensitive to climate change and human activities and would
accelerate drought expansion by the end of twenty-first century (Huang et al., 2016).
Eight sites over East Asian region are labeled with red colors, and five sites over
Central Asian area are labeled with blue colors. The major Great deserts or Gobi
deserts along with plateaus are marked with black font (e.g., Great Gobi desert in
Mongolia, Taklimakan Desert, Thar Desert, Karakum Desert, Tibetan Plateau, Loess
Plateau, and Iranian Plateau). In order to quantitatively explore detailed spectral
absorptive characteristics of dust aerosols over East and Central Asia, we choose four
East Asian sites (SACOL, Dalanzadgad, Beijing, and Yulin) and four Central Asian
sites (Dushanbe, Karachi, Kandahar, and IASBS). They consist of: SACOL located
over Loess Plateau of northwest China (Huang et al., 2008b; Guan et al., 2009; Huang
et al., 2010b; Wang et al., 2010a), Dalanzadgad in the Great Gobi of southern
Mongolia (Eck et al., 2005), Beijing in the downwind of Inner Mongolia (Xia et al.,
2007), Yulin on the southwestern fringe of the Mu Us desert in northwest China (Xu
et al., 2004; Che et al., 2009, 2015), Dushanbe in Tadzhikistan situated at the transport
corridor of Central Asian desert dust (i.e. Karakum Desert; Golitsyn and Gillette,
1993), Karachi located in the southern margin of Thar Desert in Pakistan and about 20
km from the east coast of Arabian Sea (Alam et al., 2011), Kandahar in the arid area
of southern Afghanistan, IASBS on the Iranian Plateau of northwest Iran.





### 2.2. Sun Photometer Measurements

AERONET is an internationally federated global ground-based aerosol monitoring network utilizing Cimel sun photometer, which comprises more than 500 sites all over the world (Holben et al., 1998). The Cimel Electronique sun photometer (CE-318) takes measurements of sun direct irradiances at multiple discrete channels within the spectral range of 340-1640 nm, which can be calculated aerosol optical depth (AOD) and columnar water vapor content (WVC) in centimeter. Furthermore, the instrument can perform angular distribution of sky radiances at 440, 675, 870, and 1020 nm (nominal wavelengths), which can be simultaneously retrieved aerosol volume size distribution, complex refractive index, single-scattering albedo, and asymmetry factor under cloudless condition (Dubovik and King, 2000; Dubovik et al., 2002a, 2006). The total accuracy in AOD for a newly calibrated field instrument is about 0.010-0.021 (Eck et al., 1999). The retrieval errors of SSA, ASY, Ri, and Re are anticipated to be 0.03-0.05, 0.04, 30%-50%, and 0.025-0.04, respectively, relying on aerosol types and loading (Dubovik et al., 2000). It should be borne in mind that these uncertainties are dependent on $AOD_{440} \geq 0.4$ and for solar zenith angle >50° (Level 2.0 product), and the retrieval errors will become much greater when $AOD_{440} <0.4$. The datasets of selected 13 AERONET sites in this study come from the Level2.0 product, which are pre- and post-field calibrated, automatically cloud screened, and quality-assured (Smirnov et al., 2000). In addition, a mixture of randomly oriented polydisperse spheroid particle shape assumption with a fixed aspect ratio distribution is applied to retrieve key optical properties of Asian dust (Dubovik, et al., 2002a, 2006).

### 3. Asian Dust Optical properties

A great amount of publications have verified that mineral dust aerosols are commonly predominant by large particles with coarse mode (radii>0.6 μm), which are the essential feature differentiating the dust from fine-mode dominated biomass burning and urban-industrial aerosols (Dubovik et al., 2002b; Eck et al., 2005; Bi et al., 2011, 2014; Kim et al., 2011). In other word, the values of Ångström exponent at



440-870 nm ($\alpha$) for dust aerosols usually range between -0.1 to 0.6. As pointed out by
Smirnov et al. (2002) and Dubovik et al. (2002b), sea salt aerosol is also dominant by
coarse mode and has small Ångström exponent (~0.3-0.7) but with low $AOD_{440}$
(~0.15-0.2) compared to dust aerosol. Moreover, the selected desert locations in this
study are mostly not affected by sea salt. By virtue of these differences, we can
distinguish Asian dust aerosols from other fine-mode dominated non-dust particles.
The criteria of two thresholds are put forward. (I) The particles with high aerosol
optical depth at 440 nm ($AOD_{440} \geq 0.4$) and low Ångström wavelength exponent at
440-870 nm ($\alpha < 0.2$) are defined as Pure Dust (PDU) that keep high accuracy of pure
Asian dust and eliminate most fine mode aerosols. (II) The particles with $AOD_{440} \geq 0.4$
and $0.2 < \alpha < 0.6$ are designated as Transported Anthropogenic Dust (TDU), which are
mainly dominated by dust and might mix with other anthropogenic aerosol types
during transportation. The definition of anthropogenic dust in this study is different
from earlier literatures (Tegen and Fung, 1995; Prospero et al., 2002; Huang et al.,
2015), which define that anthropogenic dust is primarily produced by various human
activities on disturbed soils (e.g., agricultural practices, industrial activity,
transportation, desertification and deforestation). It is still a huge challenge to
discriminate between natural and anthropogenic components of dust aerosols by using
current technology, AERONET products or in-situ measurements. Recently, Ginoux et
al. (2012) first estimated that anthropogenic sources globally account for 25% based
on Moderate Resolution Imaging Spectroradiometer (MODIS) Deep Blue dust optical
depth in conjunction with other land use data sets. Huang et al. (2015) proposed a new
algorithm for distinguishing anthropogenic dust from natural dust by using
Cloud-Aerosol Lidar and Infrared Pathfinder Satellite Observation (CALIPSO) and
planetary boundary layer (PBL) height retrievals along with MODIS land cover data
set. They revealed that anthropogenic dust produced by human activities mainly
comes from semi-arid and semi-humid regions and is generally mixed with other
types of aerosols within the PBL that is more spherical than natural dust. Thereby, we
assume that anthropogenic dust aerosol originated from Asian arid or semi-arid areas
has got smaller size distribution (thus larger Ångström exponent) than that of pure



natural dust.

Before insight into dust aerosol optical characteristics, we first analyze the

occurrence frequency of Asian dust over the study region that significantly affects the
intensity and distribution of mineral dust loading. Figure 2 depicts the total number
days of each month for Pure Dust ($\alpha<0.2$) and transported Anthropogenic Dust
($0.2<\alpha<0.6$) at selected four East Asian sites and four Central Asian sites. The dust
events at four East Asian sites primarily concentrate on springtime and corresponding
peak days for PDU and TDU both appear in April. This is greatly attributed to the
intrusion of dust particles during spring when dust storms are prevalent over these
regions (Wang et al., 2008). For SACOL and Beijing sites, both the PDU and TDU
days also occur in whole year except for autumn when is the rainy season, which is
linked to long-range transport of dust particulates from desert source areas and locally
anthropogenic dust (e.g., agricultural cultivation, overgrazing, desertification,
industrial and constructed dust in urbanization). Shen et al. (2016) have demonstrated
that urban fugitive dust generated by road transport and urban construction
contributes to more than 70% of particulate matter ($PM_{2.5}$) in northern China. The
dust episodes in Dushanbe of Tadzhikistan mostly happen from July to October,
which are the peak seasons of dust storms (Golitsyn and Gillette, 1993). For Karachi
site in Pakistan, the dust activities take place in spring and summer seasons. This is
because the region is not affected by the summer monsoon, leaving the land surface
sufficiently dry, and hence susceptible to wind erosion by strong winds and
meso-scale thunderstorm events typical of this time of year (Alizadeh Choobari et al.,
2014). In addition, the transport of summer dust plumes from the Arabian Peninsula
can partially contribute dust particles to Karachi site. Note that the occurred months of
PDU days are nearly different from TDU days at Dalanzadgad, Kandahar, and IASBS
sites, suggesting that dust aerosols over these areas are rarely affected by
anthropogenic pollutants. For Kandahar site in Afghanistan, the limited sampling days
to some extent may affect the statistical results. Generally, the aforementioned
occurrence frequency of dust storms over diverse sites are principally dependent on
different climatic regime and synoptic pattern, for instance, geographical location,





atmospheric circulation, wet season and dry season.
Table 1 summarizes the site information, sampling period, overall average optical
properties at 550 nm (e.g., SSA, ASY, Re, Ri, and Ångström exponent at 440-870 nm)
for Asian PDU ($\alpha<0.2$), and total number of PDU and TDU ($0.2<\alpha<0.6$) days. Note
that dust optical feature at a common 550 nm wavelength is utilized here, which can
be derived from logarithmic interpolation between 440 and 675 nm. It is worth
pointing out that the absorption and optical properties of dust aerosols at two
Dunhuang sites exhibit consistent features despite of different sampling periods,
which indicate that the chemical composition of dust aerosol at Dunhuang area
remains relatively stable.
The SSA or Ri of complex refractive index can characterize the absorptive
intensity of dust aerosols, and determine the sign (cooling or heating, depending on
the planetary albedo) of the radiative forcing (Hansen et al., 1997). Both two
quantities are mainly relied on the ferric oxide content in mineral dust (Sokolik and
Toon, 1999). Figure 3 illustrates the overall average spectral behavior of key optical
properties for PDU ($\alpha<0.2$) and TDU ($0.2<\alpha<0.6$) at selected four East Asian sites.
The SSA, ASY, Re and Ri of complex refractive index as a function of wavelength
(440, 675, 870, and 1020 nm) are presented. For all cases, the spectral behaviors of
aerosol optical parameters exhibit similar features, which can be representative of
typical patterns of Asian dust. The SSA values systematically increase with
wavelength at 440-675 nm and keep almost neutral or slight increase for the
wavelengths greater than 675 nm, which is consistent with the previous results of dust
aerosols (Dubovik et al., 2002b; Eck et al., 2005; Bi et al., 2011). In contrast, an
opposite pattern is displayed by imaginary part of refractive index, namely, Ri values
dramatically decrease from 440 nm to 675 nm, and preserve invariant from 675 nm to
1020 nm. These variations indicate that Asian dust aerosols have got much stronger
absorptive ability at shorter wavelength. Alfaro et al. (2004) implied that the
absorption capacity of soil dust increase linearly with iron oxide content, and
estimated SSA at 325 nm (~0.80) is much lower than at 660 nm (~0.95). Sokolik and
Toon (1999) revealed that ferric iron oxides (e.g., hematite and goethite) are often




internally mixed with clay minerals and result in significant dust absorption in the
UV/visible wavelengths. Hence, the spectral variations of SSA and Ri with
wavelengths are attributable to the domination of coarse-mode dust particles that have
larger light absorption in the blue spectral band as mentioned above. It is worth noting
that spectral ASY values remarkably reduce from 440 nm to 675 nm, and are almost
constant at 675-1020 nm range. This suggests that Asian dust aerosols have much
stronger scattering at 440 nm than other longer visible wavebands, due to the
contribution of coarse mode particles. By contrast, the spectral behavior of Re is not
obvious for PDU and TDU at all sites, and the mean Re values at 440 nm vary
between 1.50 and 1.56. Although there are 18 years continuous AERONET datasets at
Dalanzadgad site, the effective days of PDU and TDU are only 8 and 6 days,
respectively, almost appearing in springtime period. There are no identifiable
differences for dust absorption properties between PDU and TDU cases for
Dalanzadgad, which indicates again that the site is hardly influenced by
anthropogenic pollutants. The spectral discrepancies of optical characteristics between
PDU and TDU at other three sites show much more apparent than Dalanzadgad,
which is ascribed to these regions are not only affected by dust aerosols, but also
including local anthropogenic emissions, for instance, urban-industry, coal fuel
combustion, biomass burning, mobile source emissions, and agricultural dust (Xu et
al., 2004; Xia et al., 2007; Che et al., 2015; Bi et al., 2011; Wang et al., 2015).

Figure 4 is the same as Figure 3, but for selected four Central Asian sites. The

wavelength dependencies of PDU and TDU cases at Central Asian sites are consonant
with that of East Asian sites, despite of somewhat different variations of magnitude
and amplitude. This is expected, because the East Asian desert sites are very close to
the Central Asian desert locations and remain similar chemical compositions of dust
aerosols (Wang et al., 2004). The spectral behaviors of dust optical properties for
PDU at Kandahar and IASBS sites are nearly the same as TDU cases, which agrees
well with the consistent variability of occurrence of dust storms. The wavelength
dependency of dust characteristics for PDU at Dushanbe and Karachi presents large
differences with TDU case, which is also likely due to the influence of local



anthropogenic pollutions. Furthermore, the standard deviation of PDU is far less than
that of TDU at all wavelengths, suggesting that the robustness of PDU recognition
method.
Particle size distribution is another critical agent for deciding the optical and
radiative properties of dust aerosol. Nakajima et al. (1996) and Dubovik and King
(2000) uncovered that based on the spherical Mie theory, the retrieval errors of
volume size distribution do not exceed 10% for intermediate particle size ($0.1 \leq r \leq 7$
μm) and may greatly increase to 35-100% at the edges of size range (r<0.1 μm or r>7
μm). As mentioned above, a polydisperse, randomly oriented spheroid method is
utilized in this study, which is demonstrated to remove the artificially increased size
distribution of fine particle mode with $AOD_{440} \geq 0.4$ and for solar zenith angle >50°.
Additionally, the large errors at the edges do not significantly affect the derivation of
the main features of the particle size distribution (concentration, median and effective
radii, etc.), because typical dust aerosol size distributions have low values at the edges
of retrieval size interval (Dubovik et al., 2002a). Figure 5 delineates the overall
average columnar aerosol volume size distributions (dV/dlnr, 0.05 μm$\leq$r$\leq$15 μm) for
Pure Dust (α<0.2) and Transported Anthropogenic Dust (0.2<α<0.6) at selected 13
AERONET sites. Corresponding $AOD_{440}$ and effective radius of coarse mode ($r_{coarse}$)
in μm are also shown. It is apparent that the dV/dlnr exhibits a typical bimodal
structure and is dominant by coarse mode for PDU and TDU at all sites. The dV/dlnr
peak of coarse mode particle varies dramatically and appears at a radius $r_{Vc} \sim 2.24$ μm
for all PDU and TDU cases, while the corresponding peak of fine mode particle arises
at a radius $r_{Vf} \sim 0.09$-0.12 μm. The dV/dlnr peak and effective radius ($r_{coarse}$) of coarse
mode particles strikingly increase with the increase of AOD ascribed to the intrusion
of dust particles. For instance, the $AOD_{440}$, dV/dlnr peak values of coarse mode, and
$r_{coarse}$ for PDU at Minqin site are 0.48, 0.31 μm$^3$/μm$^2$, and 1.74 μm, respectively, and
corresponding values are 1.13, 0.77 μm$^3$/μm$^2$, and 1.93 μm at Lahore site, as shown
in Fig. 5(a). The average volume median radii of fine-mode and coarse-mode particles
for PDU are 0.159 μm and 2.157 μm, respectively, and 0.140 μm and 2.267 μm for
TDU (see Table. 2). The mean volume concentration ratio of coarse mode to fine





mode particles ($C_{vc}/C_{vf}$) for Pure Dust is about 18 (varying between 11~31) over East

and Central Asia, which is close to the average of ~20 at Dunhuang_LZU during the

spring of 2012 (Bi et al., 2014), and much less than that over Saharan pure desert

domain (~50) (Dubovik et al., 2002b). The dV/dlnr peak of coarse mode for TDU is

clearly smaller than that for PDU, and corresponding mean $C_{vc}/C_{vf}$ value is 9 (~5-11).

We attribute the high fractions of coarse-mode particles to high AOD and low

Ångström exponent values.

In this paper, we postulate that Asian dust particles only possess scattering and

absorption characteristics. And the absorption AOD value (AAOD) at a specific

wavelength can be obtained from SSA and AOD, namely, $AAOD_\lambda=(1-SSA_\lambda)\times AOD_\lambda$,

where $\lambda$ is the wavelength. Thereby, the corresponding absorption Ångström exponent

at 440-870 nm (AAE) is calculated from spectral AAOD values by using a log-linear

fitting algorithm. Figure 6 outlines the total average Ångström exponent ($\alpha$) and

absorption Ångström exponent at 440-870 nm, volume concentration of coarse mode

in $\mu m^3/\mu m^2$, and volume median radius of coarse mode in $\mu m$ for TDU ($0.2<\alpha<0.6$)

and PDU ($\alpha<0.2$) at selected AERONET sites. There are very big differences of all

quantities between PDU and TDU cases, except for some sites (e.g., Dunhuang and

Minqin). The primary reason is that we only acquire limited datasets of dust days

during spring time at Dunhuang and Minqin sites, which are hardly affected by other

anthropogenic pollutants. The AE values of TDU show remarkable changes among

each site, ranging from 0.24 to 0.44, whereas corresponding values of PDU keep

comparatively slight variations for selected 13 sites (~0.04-0.15). Furthermore, all the

AAE values of PDU are greater than 1.5, ranging between 1.65 and 2.36, and the

AAE of TDU vary from 1.2 to 2.3. We can conclude that the Asian pure dust aerosols

have got AE values smaller than 0.2 and corresponding AAE larger than 1.50, which

is another typical feature distinguishing with other non-dust aerosols. Yang et al.

(2009) attributed the high AAE values of dust aerosol in China to the presence of

ferric oxides. It is evident that volume concentrations of coarse mode for PDU are

significantly higher than TDU case, which is expected for the more coarse-mode

particles in PDU. While the volume median radius of coarse mode for TDU is greater





than PDU case, although there are some smaller values for TDU at Dalanzadgad and
Yulin sites. This is owing to dust particles at these sites usually mix with other
anthropogenic aerosol species and substantially enhance their median radii.
Figure 7 characterizes the overall mean optical properties (e.g., SSA, ASY, Re,
and Ri) at 440 nm for selected 13 sites. In general, the absorption capacity of PDU is
less than that for TDU. That is, higher SSA and smaller Ri values for PDU, except for
Dalanzadgad site. A reasonable interpretation is that threshold criterion method for
PDU in this study has effectively eliminated the fine mode aerosols, which are mostly
the much stronger absorbing aerosols (e.g., soot and biomass burning aerosol) over
East and Central Asia but weaker absorbing pollution aerosols (i.e., sulfate and nitrate)
over Dalanzadgad. Wu et al. (2012, 2014) have documented that sulfate and nitrate in
background atmosphere most likely originated directly from surface soil at the north
and south edges of Taklimakan desert and comprised steadily about 4% of dust
particulate matters, which could partially explain our results. Additionally, the overall
mean ASY and Re of PDU are greater than that of TDU, which again verifies that the
Asian pure dust has got much stronger forward scattering ability than the mixture of
Asian dust. Note that the standard deviation of SSA and Ri for PDU is a factor of two
to four lower than those from TDU. And the total average values of SSA, ASY, Re,
and Ri at 550 nm wavelength for Asian PDU are 0.935±0.014, 0.742±0.008,
1.526±0.029, and 0.00226±0.00056, respectively, while corresponding values are
0.921±0.021, 0.723±0.009, 1.521±0.025, 0.00364±0.0014 for TDU. Yang et al. (2009)
took advantage of various in situ aerosol optical and chemical measurements at
Xianghe, China during the EAST-AIRC campaign, and deduced a refractive index of
1.53-0.0023i at 550 nm of dust aerosol, which is close to the result of PDU in this
study. Nevertheless, the TDU case should be much closer to actual airborne dust
aerosol in the real world. When the elevated dusts over desert source regions are
transported eastward, they generally mix with other chemical species and react
heterogeneously with anthropogenic pollutants, and thus may significantly modify
their chemical composition and microphysical properties (Arimoto et al., 2004).
Recently, Kim et al. (2011) presented that the annual mean SSA, ASY, Re, and Ri of





complex refractive index for nearly pure Saharan dust are 0.944±0.005, 0.752±0.014,
1.498±0.032, and 0.0024±0.0034 at 550 nm, respectively, which are close to our
results of pure Asian dust but exist some differences of quantitative values and
spectral behaviors.
Average spectral optical properties (at 440, 675, 870, and 1020 nm) for PDU and
TDU over East and Central Asian regions are tabulated in Table 2. To our knowledge,
this is the first built on Asian dust optical characteristics utilizing multiyear and
multi-site AERONET measurements, which will hopefully improve uncertainties of
Asian dust shortwave radiative forcing in current regional and global climate models.
**4. Discussion**
Figure 8 describes the mean spectral behaviors of Re, RI, and SSA for Asian Pure
Dust ($\alpha$<0.2) in this study along with published dust results over various geographical
locations (Carlson and Caverly, 1977 or C77; Patterson et al., 1977 or P77; WMO,
1983; Hess et al., 1998 or OPAC; Dubovik et al., 2002b or Persian Gulf; Alfaro et al.,
2004 or Ulan Buh Desert; Wang et al., 2004 or ADEC; Todd et al., 2007 or T07). It is
well known that a lot of present-day dust models commonly take advantage of the
Optical Properties of Aerosols and Clouds (OPAC, Hess et al., 1998) or World
Meteorological Organization (WMO, 1983) databases. Curves C77 and P77 show the
complex refractive index of Saharan dust in Cape Verde Islands, Barbados West
Indies, Tenerife Canary Islands obtained from laboratory analysis by Carlson and
Caverly (1977) and Patterson et al. (1977), respectively. Curve P77 gives one of the
most widely used datasets of Ri value in the range 300-700 nm. Curve Persian
Gulf(98-00) displays the refractive index and SSA of dust over Bahrain-Persian Gulf
Desert during period of 1998-2000 derived from Dubovik et al. (2002b). Curve T07
shows the optical properties of mineral dust over Bodélé Depression of northern Chad
during 2005 retrieved from Cimel sun photometer by Todd et al. (2007). And the
curves ADEC and Ulan Buh exhibit the dust absorptive properties over
aforementioned Taklimakan Desert and Ulan Buh Desert of northwest China by Wang
et al. (2004) and Alfaro et al. (2004). Figure 8(a) presents that the spectral behaviors





of Re have relatively slight variations with values ranging from 1.50-1.56 apart from
T07 that shows lower Re values of 1.44-1.47. Todd et al. (2007) utilized Scanning
Electron Microscope (SEM) analysis of airborne dust material and confirmed that the
mineral dust is dominated by fragmented fossil diatoms from the dry lake bed of the
Bodélé Depression, which is to some extent different from the typical desert soil. As
shown in Figure 8(b), wavelength dependences of Ri exhibit comparably greater
differences in UV wavebands. In mid-visible and near infrared, our results are slightly
larger than Persian Gulf (98-00) and T07 that are retrieved from Cimel sun
photometer, but still comparable. It is very distinct that the absorbing ability of Asian
pure dust ($\alpha$<0.2) in the whole spectrum range is about a factor of 4 smaller than
current dust models (WMO, 1983; Hess et al., 1998), and is a factor of 2 to 3 lower
than the results from in situ measurements combined with laboratory analysis or
model calculations (Carlson and Caverly, 1977; Patterson et al., 1977; Wang et al.,
2004). Meanwhile, the wavelength dependences of SSA agree well with Persian Gulf
(98-00) and Ulan Buh Desert, but are much higher than OPAC. The discrepancy
increases dramatically with decreasing wavelength. Such big differences of dust
absorption capacity for diverse dust models (OPAC and WMO) and researches will
certainly lead to different radiative impacts on regional or global climate change.
Figure 9 draws the aerosol shortwave direct radiative effects (ARF) at the top of
atmosphere (TOA), at the surface (SFC), and in the atmospheric layer (ATM) for
Asian Pure Dust ($\alpha$<0.2) and Transported Anthropogenic Dust (0.2<$\alpha$<0.6) acquired
in this study, and corresponding ARF values for OPAC Mineral accumulated (Mineral
acc.) and transported (Mineral tran.) modes are also presented for comparison. We
make use of the Santa Barbara Discrete-ordinate Atmospheric Radiative Transfer
model (SBDART, Ricchiazzi et al., 1998) to calculate the ARF, which has been
proved to be a reliable software code and widely used for simulating plane-parallel
radiative fluxes in the Earth's atmosphere (Halthore et al., 2005; Bi et al., 2013). The
main input parameters of spectral AOD, surface albedo, WVC, and columnar ozone
amount are prescribed to same values, and the spectral SSA, ASY, Re, and Ri values
are obtained from aforementioned various dust models. It is evident that Earth's



energy budget is modulated and redistributed by different absorbing properties of
mineral dusts. The results indicate that the cooling rate at SFC (negative radiative
forcing) gradually increases with PDU ($\alpha<0.2$), TDU ($0.2<\alpha<0.6$), OPAC Mineral
accumulated and transported dust modes. By contrast, the cooling intensity at TOA
gradually decreases with diverse dust cases, and even becomes positive radiative
forcing for OPAC transported dust mode, with ARF varying from -15.6, -13.8, -6.9,
and +0.24 $\text{Wm}^{-2}$, respectively. Therefore, the heating intensity in the atmospheric
layer sharply increases from +22.7, +29.5, +46.6, and +58.3 $\text{Wm}^{-2}$. The heating rate in
ATM for OPAC Mineral (acc. and tran.) modes is about two-fold greater than Asian
dust cases (PDU and TDU). Such large diabatic heating rates might warm the dust
layer, suppress the development of convection under the lower atmosphere, thus exert
profound impacts on the atmospheric dynamical and thermodynamic structures and
cloud formation together with the strength and occurrence frequency of precipitation
(Rosenfeld et al., 2001; Huang et al., 2010a; Creamean et al., 2013). Hence, accurate
and reliable absorbing characteristics of Asian dust should be considered in
present-day regional climate models.
**5. Summary**
In this study, we have proposed two threshold criteria to discriminate two types of
Asian dust: Pure Dust (PDU, $\alpha<0.2$) and Transported Anthropogenic Dust (TDU,
$0.2<\alpha<0.6$). PUD can represent nearly "pure" dust in desert source regions and
decrease disturbance of other non-dust aerosols, which would also exclude some fine
mode of dust particles. The spectral behaviors of TDU exhibit similar variations with
PDU, but show much stronger absorption and weaker scattering than PDU cases.
There are two markedly identifiable characteristics for Asian PDU. (I) spectral SSA
values systematically increase with wavelength from 440 nm to 675 nm and remain
almost neutral or slight increase for the wavelength greater than 675 nm, whereas an
opposite pattern is shown for imaginary part of refractive index. (II) Asian pure dust
aerosols have got AE values smaller than 0.2 and AAE larger than 1.50. Compared
with current common dust models (e.g., OPAC and WMO), Asian dust aerosol has



relatively weak absorption for wavelengths greater than 550 nm (SSA~0.96-0.99), but
presents a moderate absorption in the blue spectral range ($SSA_{440}$~0.92-0.93). The
overall average values of SSA, ASY, Re, and Ri at 550 nm wavelength for Asian PDU
are 0.935±0.014, 0.742±0.008, 1.526±0.029, and 0.00226±0.00056, respectively,
while corresponding values are 0.921±0.021, 0.723±0.009, 1.521±0.025,
0.00364±0.0014 for TDU.
It should be noted that the definition of anthropogenic dust in this paper is
ambiguous, and TDU here represents more accurately dominant dust mixing with
other anthropogenic aerosols. Because it is very difficult to quantify the
anthropogenic contribution due to large uncertainties in defining the anthropogenic
fraction of ambient dust burden (Sokolik et al., 2001; Huang et al., 2015). Diverse
human activities (e.g., agricultural cultivation, desertification, industrial activity,
transportation, and construction in urbanization) in vulnerable environments might
modify the land use and Earth's surface cover, and would affect the occurred
frequency and intensity of anthropogenic dust. Hence, the optical features of
anthropogenic dust aerosols are dependent on the source regions and chemical
compositions. However, as concluded by Huang et al. (2015), anthropogenic dust
generated by human activities mainly comes from semi-arid and semi-humid regions
(Guan et al., 2016) and is generally mixed with other types of aerosols within the PBL.
And we primarily investigated dust aerosols in arid or semi-arid regions over East and
Central Asia, where are somewhat disturbed by human activities. Therefore, the key
optical properties of TDU derived from this study should to some extent contain the
anthropogenic fraction. To fully elucidate exact optical properties of anthropogenic
dust, we need to explore detailed morphology, mineralogy, and chemical
compositions by means of in situ measurements, laboratory analysis, active and
passive remote sensing methods (e.g., multi-wavelength lidar, AEROENT, MODIS)
as well as model calculations.

*Acknowledgements*. This work was jointly supported by the National Science Foundation of
China (41521004, 41305025, 41575015 and 41405113), the Fundamental Research Funds for the





Central Universities lzujbky-2015-4 and lzujbky-2013-ct05, and the China 111 Project (No. B
13045). We thank the GSFC/NASA AERONET group for processing the AERONET data
(http://aeronet.gsfc.nasa.gov). The authors would like to express special thanks to the
principal investigators (Hong-Bin Chen, Philippe Goloub, Bernadette Chatenet, Xiao-Ye Zhang,
Laurent Gomes, Sabur F. Abdullaev, and Hamid Khalesifard) and their staff for effort in
establishing and maintaining the instruments at AERONET sites used in this work. We appreciate
the MODIS and TOMS teams for supplying the satellite data. We would also like to thank all
anonymous reviewers for their constructive and insightful comments.

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



**Figure captions**

**Table 1.** Overall average and standard deviation of key optical properties at 550 nm (e.g.,
single-scattering albedo, asymmetry factor, real part and imaginary part of complex refractive
index) for Asian pure Dust (PDU). Ångström wavelength exponent ($\alpha$) is in the range of 440-870
nm. Minimum and maximum values of the optical properties are in parenthesis for each
corresponding column. Measuring period and the total number of PDU ($\alpha<0.2$) and Transported
Anthropogenic Dust (TDU, $0.2<\alpha<0.6$) days are in the parenthesis for the first and last column,
respectively.

| Site (sampled period) | SSA (min, max) | ASY (min, max) | Re (min, max) | Ri ($\times10^{-3}$) | Ångström (440-870 nm) | PDU/days (TDU) |
|---|---|---|---|---|---|---|
| SACOL | 0.932±0.018 | 0.741±0.012 | 1.534±0.044 | 2.251±0.788 | 0.120±0.049 | 38 |
| (2006-2012) | (0.888, 0.971) | (0.715, 0.771) | (1.438, 1.60) | (0.913, 5.51) | (0.0, 0.198) | (97) |
| Dalanzadgad | 0.930±0.012 | 0.746±0.010 | 1.512±0.046 | 2.407±0.414 | 0.127±0.079 | 8 |
| (1997-2014) | (0.912, 0.949) | (0.724, 0.766) | (1.447, 1.60) | (1.649, 3.19) | (-0.06, 0.199) | (6) |
| Beijing | 0.917±0.020 | 0.742±0.012 | 1.557±0.043 | 2.801±0.865 | 0.117±0.067 | 46 |
| (2001-2015) | (0.863, 0.963) | (0.716, 0.769) | (1.401, 1.60) | (1.032, 6.20) | (-0.048, 0.199) | (67) |
| Yulin | 0.907±0.024 | 0.748±0.010 | 1.559±0.038 | 3.564±1.589 | 0.077±0.068 | 13 |
| (2001-2002) | (0.863, 0.952) | (0.731, 0.771) | (1.476, 1.60) | (1.370, 7.92) | (-0.024, 0.188) | (16) |
| Dushanbe | 0.941±0.012 | 0.739±0.011 | 1.529±0.041 | 2.011±0.551 | 0.128±0.054 | 26 |
| (2010-2015) | (0.916, 0.959) | (0.710, 0.765) | (1.436, 1.60) | (1.022, 3.475) | (-0.02, 0.198) | (95) |
| Karachi | 0.945±0.012 | 0.741±0.011 | 1.518±0.030 | 1.938±0.561 | 0.141±0.041 | 83 |
| (2006-2014) | (0.916, 0.977) | (0.714, 0.767) | (1.449, 1.60) | (0.758, 3.439) | (-0.005, 0.20) | (286) |
| Lahore | 0.930±0.014 | 0.740±0.010 | 1.519±0.038 | 2.253±0.611 | 0.136±0.052 | 26 |
| (2007-2015) | (0.901, 0.957) | (0.721, 0.765) | (1.432, 1.60) | (1.207, 3.623) | (0.023, 0.198) | (248) |
| IASBS | 0.933±0.017 | 0.725±0.011 | 1.572±0.024 | 2.290±0.845 | 0.098±0.050 | 19 |
| (2010-2013) | (0.883, 0.958) | (0.704, 0.746) | (1.525, 1.60) | (1.245, 5.029) | (0.021, 0.195) | (12) |
| Kandahar | 0.925±0.013 | 0.729±0.017 | 1.534±0.035 | 2.855±0.775 | 0.147±0.054 | 10 |
| (2008/04-06) | (0.903, 0.955) | (0.700, 0.768) | (1.492, 1.60) | (1.445, 4.65) | (0.00, 0.199) | (4) |
| Dunhuang | 0.947±0.015 | 0.745±0.013 | 1.547±0.037 | 1.714±0.697 | 0.039±0.029 | 6 |
| (2001/03-05) | (0.918, 0.970) | (0.723, 0.761) | (1.494, 1.60) | (1.014, 3.14) | (-0.003, 0.091) | (0) |
| Dunhuang_LZU | 0.958±0.007 | 0.741±0.021 | 1.495±0.042 | 1.589±0.292 | 0.153±0.026 | 5 |
| (2012/04-05) | (0.951, 0.968) | (0.707, 0.771) | (1.451, 1.580) | (1.092, 1.84) | (0.117, 0.184) | (4) |
| Inner_Mongolia | 0.948±0.012 | 0.751±0.006 | 1.499±0.042 | 1.641±0.457 | 0.069±0.054 | 4 |
| (2001/04-05) | (0.930, 0.960) | (0.743, 0.759) | (1.426, 1.54) | (1.169, 2.45) | (0.011, 0.165) | (1) |
| Minqin | 0.945±0.002 | 0.756±0.014 | 1.469±0.023 | 2.036±0.220 | 0.119±0.023 | 2 |
| (2010/05-06) | (0.942, 0.947) | (0.740, 0.764) | (1.449, 1.494) | (1.883, 2.29) | (0.103, 0.146) | (0) |
| **Overall Mean** | **0.935±0.014** | **0.742±0.008** | **1.526±0.029** | **2.258±0.556** | **0.113±0.033** | **PDU** |
| **Overall Mean** | **0.921±0.021** | **0.723±0.009** | **1.521±0.025** | **3.643±1.372** | **0.355±0.06** | **TDU** |






**Table 2.** Spectral optical properties of Pure Dust (α<0.2) and Transported Anthropogenic Dust
(0.2<α<0.6) averaged for 13 sites over East and Central Asia areas.

| Asian Dust | Pure Dust (α<0.2) | Transported Anthropogenic Dust (0.2<α<0.6) |
|---|---|---|
| $\omega_0$(440/675/870/1020) | 0.906/0.962/0.971/0.975 ±0.009 | 0.897/0.943/0.954/0.959 ±0.019 |
| Re(440/675/870/1020) | 1.520/1.533/1.517/1.503 ±0.026 | 1.509/1.533/1.532/1.525 ±0.027 |
| Ri(440/675/870/1020) $\times 10^{-3}$ | 3.413/1.574/1.449/1.449 ±0.450 | 5.064/2.737/2.510/2.486 ±1.300 |
| ASY(440/675/870/1020) | 0.758/0.727/0.724/0.726 ±0.008 | 0.736/0.711/0.710/0.712 ±0.009 |
| $r_{Vf}$ (μm); $\sigma_f$ | 0.159±0.029 | 0.140±0.011 |
| $r_{Vc}$ (μm); $\sigma_c$ | 2.157±0.112 | 2.267±0.214 |
| Cvf ($\mu m^3/\mu m^2$) | 0.037±0.011; 0.06×τ(1020)-0.001 | 0.038±0.011; 0.12×τ(1020)-0.014 |
| Cvc($\mu m^3/\mu m^2$) | 0.632±0.167; 0.88×τ(1020)-0.07 | 0.343±0.084; 0.90×τ(1020)-0.06 |
| Cvc/Cvf | 17.9 (11~31) | 9.1 (5~11) |

Each variable is accompanied by a standard deviation (e.g.., ±0.01). $r_{Vf}$ and $r_{Vc}$ are the volume
median radii of fine-mode and coarse-mode particles in μm; Cvf and Cvc denote the volume
concentrations of fine-mode and coarse-mode particles in $\mu m^3/\mu m^2$, respectively. The dynamic
dependencies of dust optical properties are exhibited as functions of $AOD_{1020}$, with correlation
coefficients greater than 0.93 for all cases.

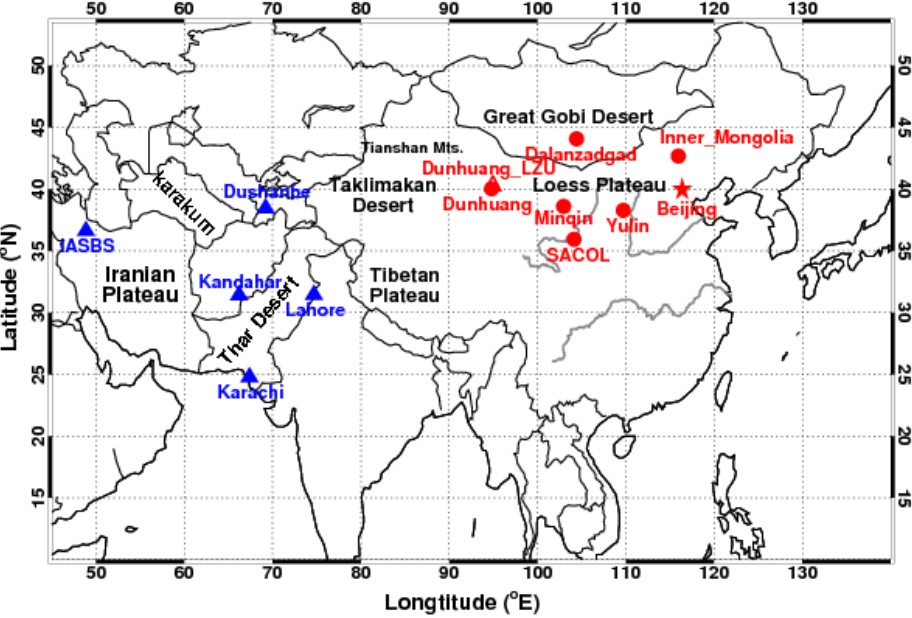


**Figure 1.** Geographical location of selected 13 AERONET sites in this study. Eight sites over East
Asian region are labeled with red colors, and five sites over Central Asian region are labeled with
blue colors. The major Great deserts or Gobi deserts along with plateaus are marked with black
font.

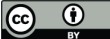




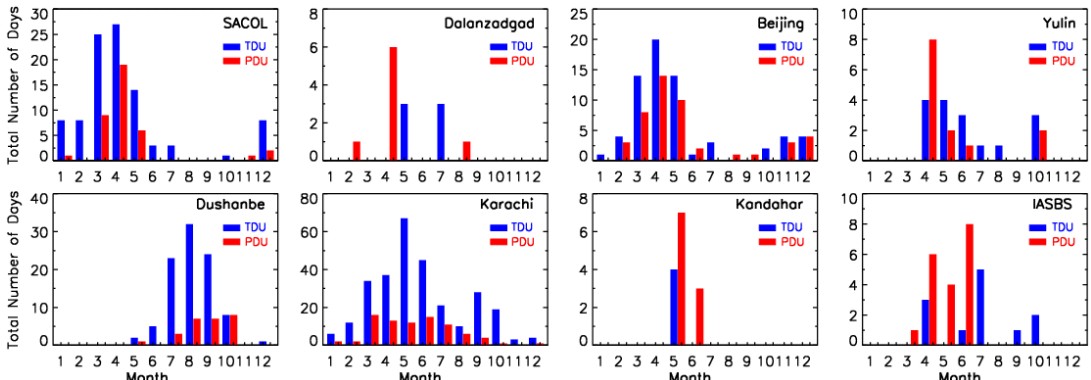


**Figure 2.** Occurrence frequency of total number days for Pure Dust (α<0.2, PDU with red color)
and Transported Anthropogenic Dust (0.2<α<0.6, TDU with blue color) at selected four East
Asian sites (top panel) and four Central Asian sites (bottom panel).


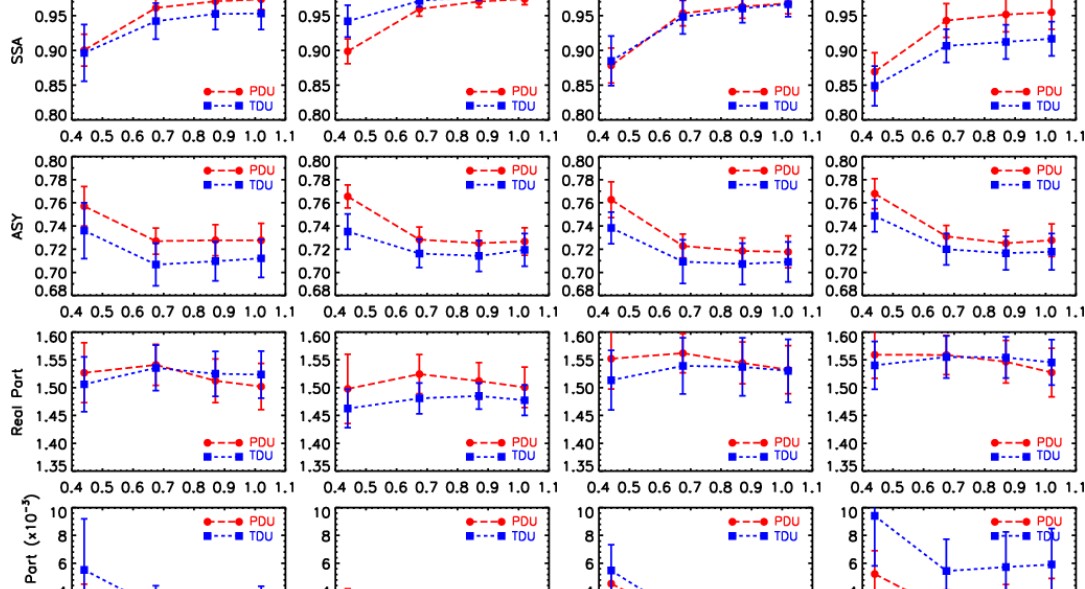


**Figure 3.** Overall average spectral behavior of key optical properties for Pure Dust (α<0.2, PDU
with red circle) and Transported Anthropogenic Dust (0.2<α<0.6, TDU with blue square) at
selected four East Asian sites (SACOL, Dalanzadgad, Beijing and Yulin). The error bars indicate
plus or minus one standard deviation.






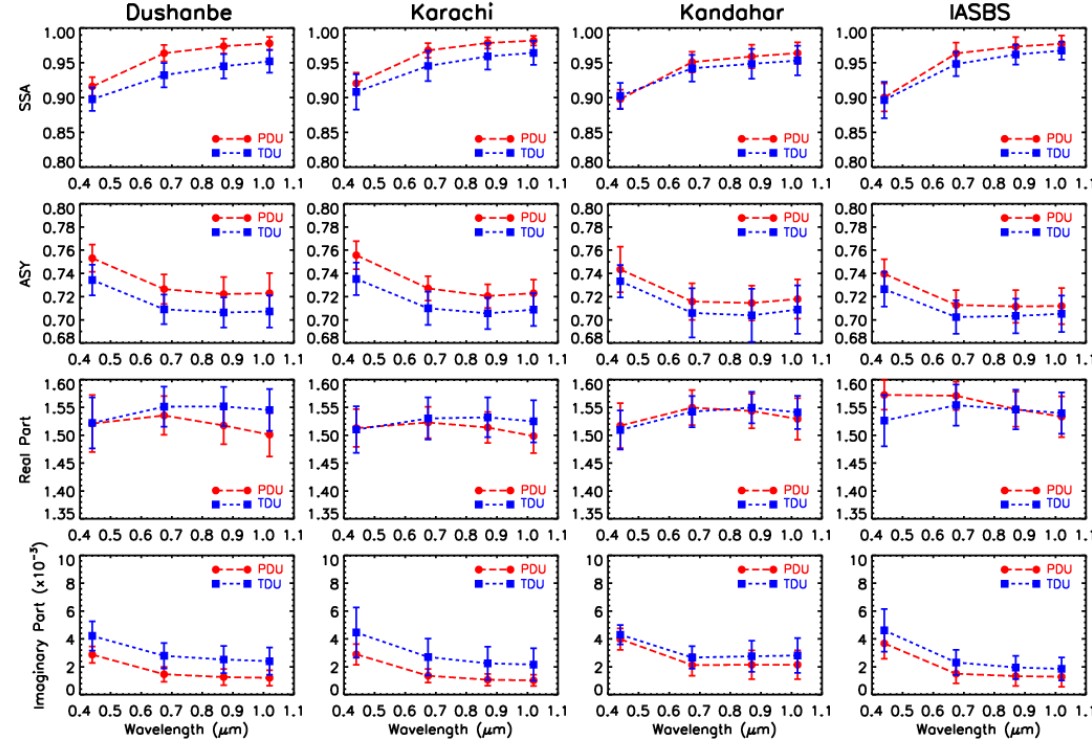


**Figure 4.** The same as Figure 3, but for selected four Central Asian sites (Dushanbe, Karachi, Kandahar and IASBS).



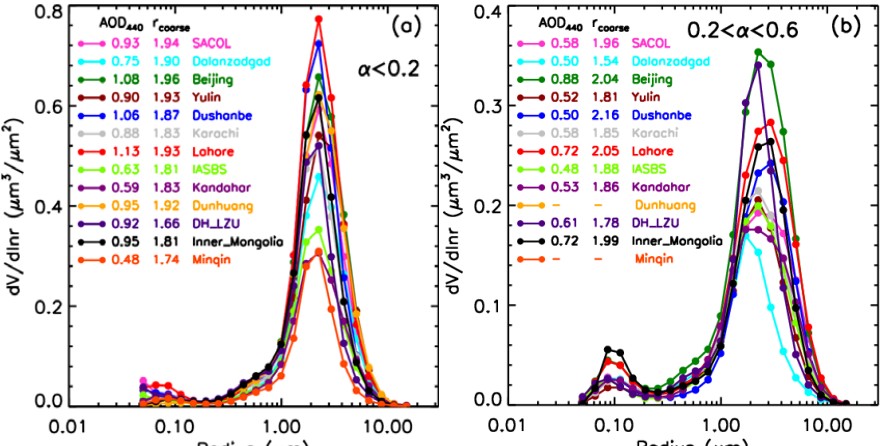


**Figure 5.** Overall average of aerosol volume size distributions in the entire atmospheric column for (a) Pure Dust (α<0.2) and (b) Transported Anthropogenic Dust (0.2<α<0.6) at selected 13 AERONET sites. Corresponding aerosol optical depth at 440 nm (AOD$_{440}$) and effective radius of




coarse mode ($r_{coarse}$) in μm are also shown.








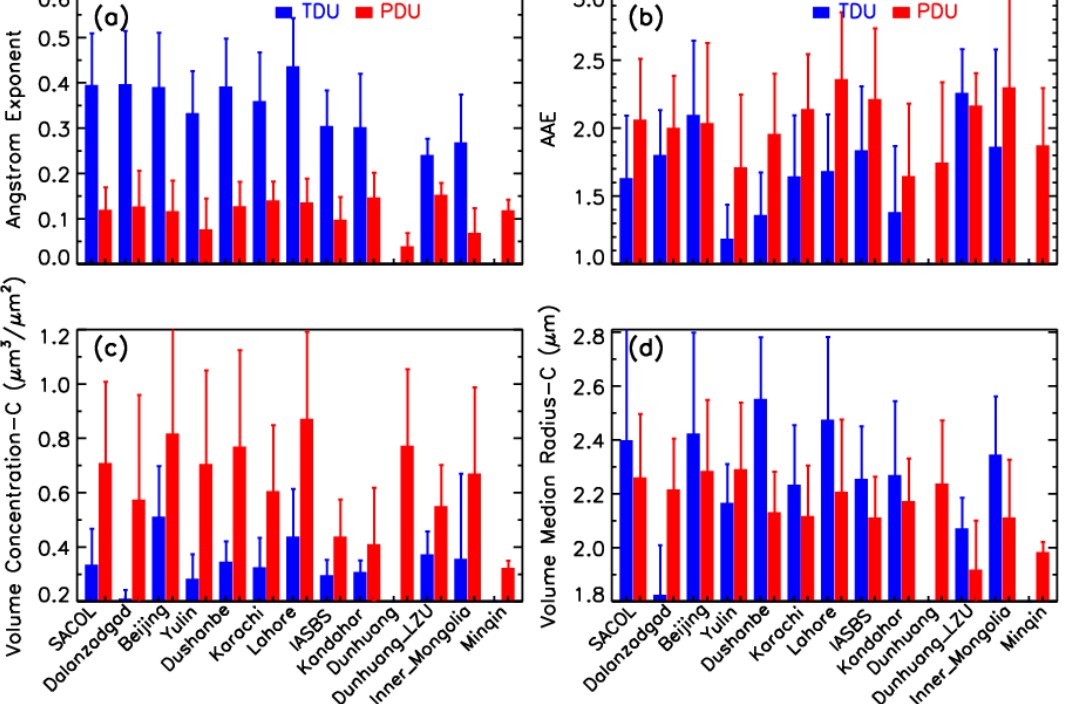


**Figure 6.** Total average values of (a) Ångström exponent (440-870 nm), (b) absorption Ångström
exponent at 440-870 nm (AAE), (c) volume concentration of coarse mode (μm³/μm²), and (d)
volume median radius of coarse mode in μm for Transported Anthropogenic Dust (0.2<α<0.6,
blue color) and Pure Dust (α<0.2, red color) at 13 selected AERONET sites. The error bars
indicate plus or minus one standard deviation.












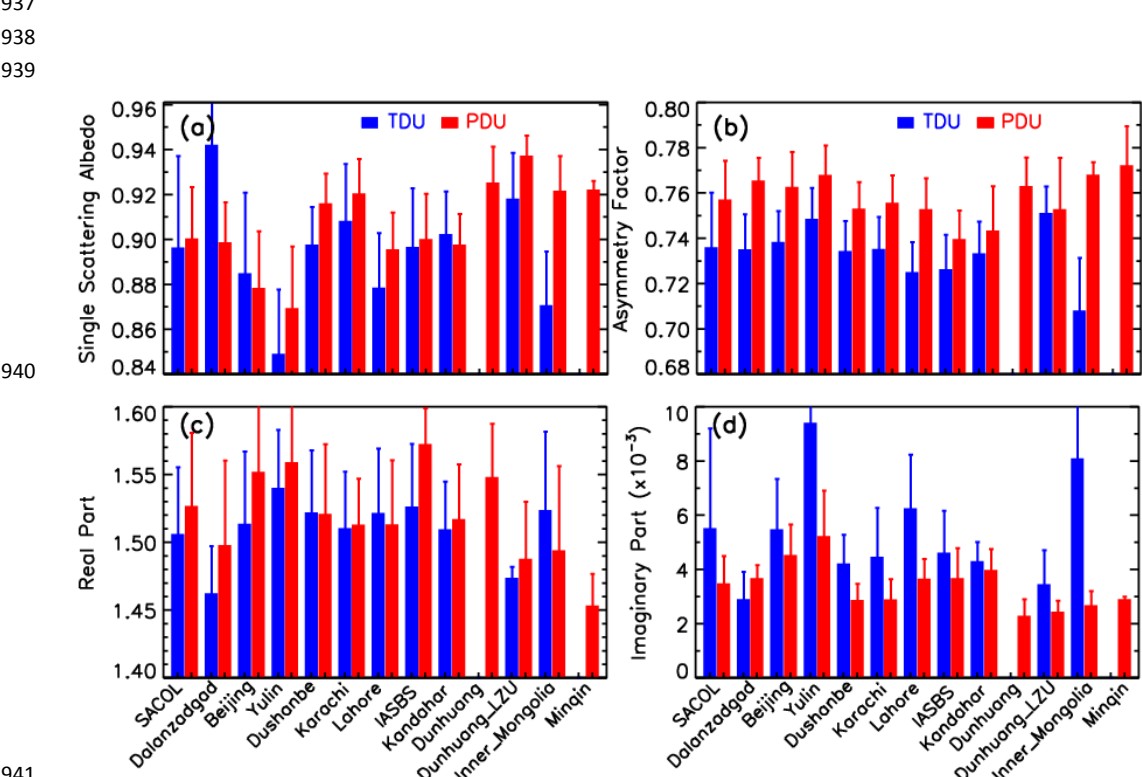


**Figure 7.** The same as Figure 5, but for (a) sing-scattering albedo, (b) asymmetry factor, (c) real
part and (d) imaginary part of complex refractive index at 440 nm.



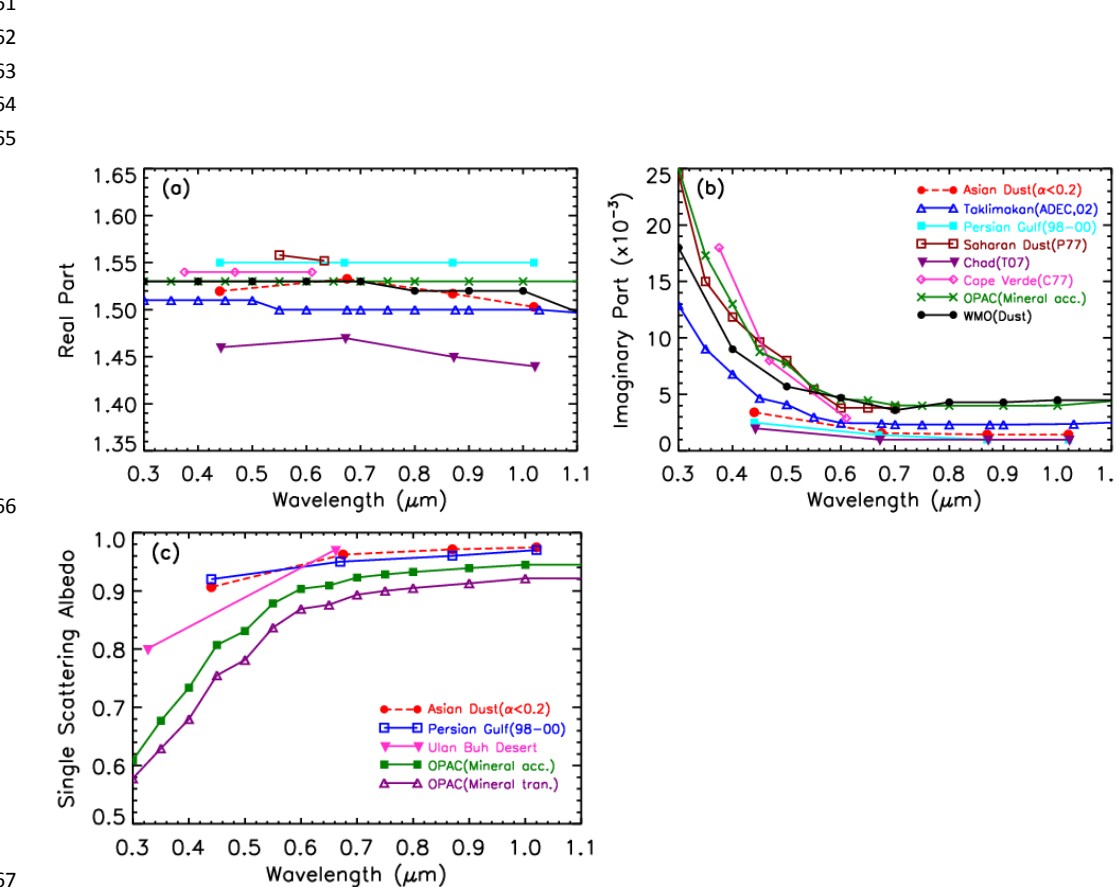



**Figure 8.** Mean spectral behaviors of (a) real part, (b) imaginary part of complex refractive index, and (c) single-scattering albedo for Asian Pure Dust ($\alpha<0.2$) calculated for 13 AERONET sites, and results of current common dust models (OPAC, WMO), Bahrain-Persian Gulf of Desert dust (1998-2000), Saharan dust (Chad, Cape Verde Islands), and Chinese Gobi desert (Taklimakan, Ulan Buh Desert) are also shown for comparison.











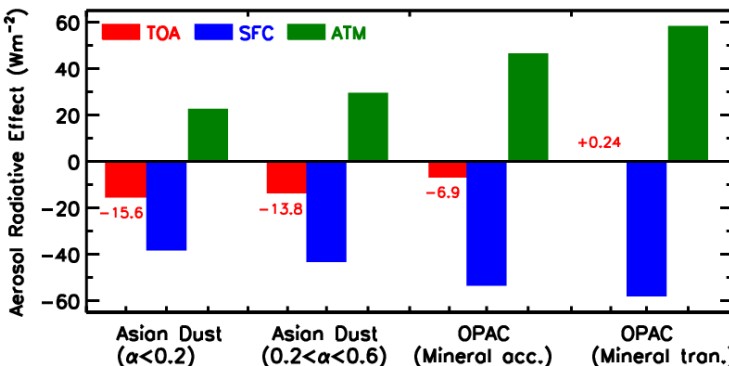


**Figure 9.** Aerosol shortwave direct radiative effects at the top of the atmosphere (TOA, red color),
at the surface (SFC, blue color), and in the atmospheric layer (ATM, green color) for Asian Pure
Dust ($\alpha<0.2$) and Transported Anthropogenic Dust ($0.2<\alpha<0.6$) computed in this study, and
corresponding values for OPAC Mineral accumulated (Mineral acc.) and transported (Mineral
tran.) modes are also presented for comparison.

993