# Peer review of "Comparison of key absorption and optical properties between pure"

_Atmospheric Chemistry and Physics, 2016_

## Referee Comment (RC1) · Anonymous Referee #1 · 13 Oct 2016

General comments: Dust aerosol is one of the major aerosol types over East and Central Asia, and the total amount and absorptive intensity of dust are crucial for determining its direct and indirect climate forcing. Thus far, the absorptive capacity of dust aerosol generated from Asian desert region is still an open question. Based on multi-year and multi-site quality assured datasets of the Aerosol Robotic Network (AERONET Level 2.0 products), the authors proposed two threshold criteria to discriminate two types of Asian dust: Pure Dust (PDU, ïĄą<ïĂřïĂőïĂš) and Transported Anthropogenic Dust (TDU, 0.2<ïĄą<ïĂřïĂőïĂű). The overall average of single-scattering albedo, asymmetry factor, real part and imaginary part of complex refractive index at 550 nm for PDU are 0.935±0.014, 0.742±0.008, 1.526±0.029, 0.00226±0.00056, re-

spectively, while corresponding values are 0.921±0.021, 0.723±0.009, 1.521±0.025, and 0.00364±0.0014 for TDU. The results of this paper are valuable and hold promise of improving accuracies of Asian dust characteristics in present-day remote sensing applications and climate models. Overall, both the English-written and grammar are well and appropriate for publication. I recommend this manuscript is accepted and published in the Journal of ACP (Atmospheric Chemistry and Physics) after minor modifications.

Suggestion: 1.How to distinguish and separate the natural and anthropogenic contributions for climate variability, has become one of the most intractable problems in current global climate change. The authors proposed two threshold criteria to identify two types of Asian dust: Pure Dust (PDU, ïĄą<ïĂřïĂőïĂš) and Transported Anthropogenic Dust (TDU, 0.2<ïĄą<ïĂřïĂőïĂű), and explore the key absorption and optical properties. These results are encouraging and helpful to update the essential parameters of Asian dust in current remote sensing applications and climate models. As mentioned in the manuscript, it is still a huge challenge to discriminate between natural and anthropogenic components of dust aerosols by using current technology, AERONET products or in-situ measurements. However, the reviewer encourages the authors to explore detailed morphology, mineralogy, and chemical compositions by means of in situ measurements, laboratory analysis, active and passive remote sensing methods (e.g., multi-wavelength lidar, AEROENT, MODIS) as well as model calculations in the future work. âĞŠãĂĂThe authors don't need to response this.

Minor comments: 1. Abstract, Page 2, line 54: "OPAC" âĞŠ Change to "Optical Properties of Aerosols and Clouds (OPAC)". When an abbreviation firstly appears in the manuscript, please give the full name.

2. Page 3, line 80: "are about 6 times larger than at 660 nm" âĞŠ Change to "are about 6 times larger than that at 660 nm"

3. Page 4, line 114: "theory calculation" âĞŠ Change to "theoretical calculation"
4. Page 8, line 209: "compared to" âĞŠ Change to "compared with"

5. Page 8, line 219: "literatures" âĞŠ Change to "literature"

6. Page 9, line 247: "linked to" âĞŠ Change to "linked with"

7. Page 17, line 494: "PUD" âĞŠ Change to "PDU"

Please also note the supplement to this comment:
http://www.atmos-chem-phys-discuss.net/acp-2016-764/acp-2016-764-RC1-
supplement.pdf

---

## Referee Comment (RC2) · Anonymous Referee #2 · 14 Oct 2016

*Comments on "Comparison of key absorption and optical properties between pure and transported anthropogenic dust over East and Central Asia." by Jianrong Bi et al.*

**General comments:**

Asian mineral dust is one of the most important aerosol species in the Earth-atmosphere system that exerts profound influences on air quality, public health, marine biogeochemical cycle and Earth's climate. However, the absorptive ability of Asian dust is still an unresolved question at present, causing large uncertainties in estimating its shortwave radiative effect in current regional models. This study compiles the key absorption and optical properties between pure dust (PDU) and transported anthropogenic dust (TDU) over East and Central Asia, by utilizing multiyear and multi-site quality assured AERONET measurements. The authors also compare the average spectral behaviors of PDU with present-day common dust models (e.g., OPAC and WMO) together with published results over various desert locations. These meaningful results are very useful to evaluate quantitatively the radiative effects of Asian dust on regional or global climate change. In general, I found the paper is well written in English, and I recommend accepted this paper for publication in the journal of Atmospheric Chemistry and Physics with minor revision.

**Specific comments:**

(1) Page 16, lines 471-473: "The main input parameters of spectral AOD, surface albedo, WVC, and columnar ozone amount are prescribed to same values (e.g., …)"

⇒Please give the prescribed values of spectral AOD, surface albedo, WVC, and columnar ozone amount in the manuscript, which would be convenient for audience.

(2) Page 33, Figure 5(b): For Minqin and Dunhuang sites, the $AOD_{440}$ and $r_{coarse}$ are shown for "-", the authors may want to indicate the missing data. Please give the

explanation in the context.

**Minor comments:**

(1) 1. Introduction, Page 5, line 142: Please add "There have been several world-famous aerosol long-term monitoring networks over Asian region for examining aerosol features and its radiative effects, for instance, AERONET—AErosol RObotic NETwork (Holben et al., 1998), SKYNET—aerosol-cloud-radiation interaction ground-based observation network (Nakajima et al., 1996; Takamura et al., 2004; Che et al., 2008), and CARSNET—China Aerosol Remote Sensing Network (Che et al., 2009, 2014, 2015)." at the beginning of this section and add corresponding cited literature in

(2) Page 7, line 200: "**3. Asian Dust Optical properties**"
⇒ Change to "**3. Asian Dust Optical Properties**"

(3) Page 7, lines 201-205: Change to "A great amount of publications have verified that mineral dust aerosols are commonly predominant by large particles with coarse mode (radii>0.6 μm), which are the essential feature differentiating the dust from fine-mode dominated biomass burning and urban-industrial aerosols (Dubovik et al., 2002b; Eck et al., 2005; Bi et al., 2011, 2014; Kim et al., 2011; Che et al., 2013)."

(4) Page 8, line 209: "compared to"
⇒ Change to "compared with"

(3) Page 8, line 219: change "literatures" to "literature" and modify the other places in the whole manuscript.

(6) Page 9, lines 259-260: "Note that the occurred months of PDU days are nearly different from TDU days at Dalanzadgad,"

⇒ Change to "Note that the occurred months of PDU cases are nearly different from TDU cases at Dalanzadgad,"

(7) Page 10, line 294: "and estimated SSA at 325 nm (~0.80) is much lower than at 660 nm (~0.95)."

⇒ Change to "and estimated SSA at 325 nm (~0.80) is much lower than that at 660 nm (~0.95)."

(8) 5. Summary, Page 17, line 494: change "PUD" to "PDU" and modify the other places in the whole manuscript.

---

## Referee Comment (RC3) · Anonymous Referee #3 · 14 Oct 2016

The work presented in this paper is very interesting and well structured. The authors suggest a method for discriminating the presence of Desert dust in the atmosphere, dividing it in two different cases: pure dust and transported anthropogenic dust. The method is based on a threshold on AOD (440 nm) and Angsrtom exponent (calculated using the two wavelengths 400 and 870 nm), and provided good results when compared with the plots of the volume size distributions. Also the section devoted to the comparison among the values retrieved from measurements and the ones from models generally used, is very interesting and useful.

To complete the paper, I suggest the authors to give a look to the following paper where a similar work has been done for Saharan dust in Europe : "Inventory of African desert

dust events over the southwestern Iberian Peninsula in 2000–2005 with an AERONET Cimel Sun photometer", Toledano et al, 2007, DOI: 10.1029/2006JD008307. Also in this paper thresholds on Angstrom exponent and AOD are used in order to set up an Automatic Criterion for Detection and Evaluation of Desert Dust Intrusions and, as expected, they are different from the ones used in this paper. I think it should be highlighted in the text that the chosen values are good for the type of dust intrusion of the selected area, and that for a smaller area or a different geographical location, they must be selected carefully. In that paper it is also written that a larger sensitivity to the presence of dust particles has been found at 870 nm rather that 440 m. Do the authors think that using a threshold on this wavelength in the case of TDU would help to discriminate more accurately the amount of dust from the anthropogenic aerosol? Did the author never found (in TDU dataset) a 3 modal volume size distribution? If yes, it could be another possibility for better understanding the composition of TDU dust.

Minor comments in addition of the ones already done by the other referees:

Line 189: put the acronyms of SSA, ASY, Ri and Re in line 186, where these quantities are listed.

192: "are dependent on AOD440>=0.4" I think it would be better saying " are valid for AOD440. . ."

277: "capability" instead of "intensity"

346: it is written that the pick radius of the coarse mode is about 2.24 for both PDU and TDU. However for Yulin in TDU it seems to be about 3. I think that in the case of TDU it would be better saying that the pick radius in between 2-3.

512-514: the sentence begins with "because" but it doesn't seems to have a correct grammatical structure ( subject, verb , object..). Please check it

Please also note the supplement to this comment:

http://www.atmos-chem-phys-discuss.net/acp-2016-764/acp-2016-764-RC3-supplement.pdf

---

## Author Comment (AC1) · 17 Oct 2016

Manuscript No.: acp-2016-764 Journal: ACP The revised manuscript entitled "Comparison of key absorption and optical properties between pure and transported anthropogenic dust over East and Central Asia." by Jianrong Bi, et al.

Response to Reviewer1: We greatly appreciated for Editor's big help! We have carefully checked and revised the manuscript according to Reviewers' comments, which are helpful and valuable for greatly improving our manuscript. Please find a point-by-point reply to the issues as follows (highlighted in blue color font). And we have also uploaded the file of "Response to-Reviewer1(acp-2016-764).pdf".

Suggestion: 1. How to distinguish and separate the natural and anthropogenic contributions for climate variability, has become one of the most intractable problems in current global climate change. The authors proposed two threshold criteria to identify two types of Asian dust: Pure Dust (PDU, ïĄą<ïĂřïĂőïĂš) and Transported Anthropogenic Dust (TDU, 0.2<ïĄą<ïĂřïĂőïĂű), and explore the key absorption and optical properties. These results are encouraging and helpful to update the essential parameters of Asian dust in current remote sensing applications and climate models. As mentioned in the manuscript, it is still a huge challenge to discriminate between natural and anthropogenic components of dust aerosols by using current technology, AERONET products or in-situ measurements. However, the reviewer encourages the authors to explore detailed morphology, mineralogy, and chemical compositions by means of in situ measurements, laboratory analysis, active and passive remote sensing methods (e.g., multi-wavelength lidar, AEROENT, MODIS) as well as model calculations in the future work. âĞŠãĂĂThe authors don't need to response this. Response: Thank you very much for Reviewer's good suggestions. To fully elucidate exact optical properties of anthropogenic dust, we shall explore detailed morphology, mineralogy, and chemical compositions by means of in situ measurements, laboratory analysis, active and passive remote sensing methods (e.g., multi-wavelength lidar, AEROENT, MODIS) as well as model calculations in our future work.

Minor comments: 1. Abstract, Page 2, line 54: "OPAC" âĞŠ Change to "Optical Properties of Aerosols and Clouds (OPAC)". When an abbreviation firstly appears in the manuscript, please give the full name. Response: We have changed "OPAC" to "Optical Properties of Aerosols and Clouds (OPAC)" in Line 54 and corresponding places in the whole text.

2. Page 3, line 80: "are about 6 times larger than at 660 nm" âĞŠ Change to "are about 6 times larger than that at 660 nm" Response: We have changed to "are about 6 times larger than that at 660 nm" in Line 80.

3. Page 4, line 114: "theory calculation" âĞŠ Change to "theoretical calculation" Response: We have changed "theory calculation" to "theoretical calculation" in Line 114.

4. Page 8, line 209: "compared to" âĞŠ Change to "compared with" Response: We have changed "compared to" to "compared with" in Line 209.

5. Page 8, line 219: "literatures" âĞŠ Change to "literature" Response: We have changed "literatures" to "literature" in Line 219.

6. Page 9, line 247: "linked to" âĞŠ Change to "linked with" Response: We have changed "linked to" to "linked with" in Line 247.

7. Page 17, line 494: "PUD" âĞŠ Change to "PDU" Response: We have changed "PUD" to "PDU" in Line 494.

Please also note the supplement to this comment:
http://www.atmos-chem-phys-discuss.net/acp-2016-764/acp-2016-764-AC1-
supplement.pdf

---

## Author Comment (AC2) · 17 Oct 2016

Manuscript No.: acp-2016-764 Journal: ACP The revised manuscript entitled "Comparison of key absorption and optical properties between pure and transported anthropogenic dust over East and Central Asia." by Jianrong Bi, et al.

Response to Reviewer2: We are grateful to the Editor and the anonymous Reviewer for their constructive and insightful comments. The comments of the Reviewers are helpful and valuable for greatly improving the manuscript. Please find a point-by-point reply to the issues as follows (highlighted in blue color font). And we have also uploaded the file of "Response to-Reviewer2(acp-2016-764)-supplement.pdf".

[Figure]

Specific comments: (1) Page 16, lines 471-473: "The main input parameters of spectral AOD, surface albedo, WVC, and columnar ozone amount are prescribed to same values (e.g., . . .)" ⇒Please give the prescribed values of spectral AOD, surface albedo, WVC, and columnar ozone amount in the manuscript, which would be convenient for audience. Response: Thank you very much for Reviewer's insightful comments. We have presented the prescribed values of spectral AOD, surface albedo, WVC, and columnar ozone amount in Lines 471-473, "(e.g., 0.72, 0.30, 1.0 cm, and 300 DU for input AOD440, surface albedo, WVC, and ozone amount)".

(2) Page 33, Figure 5(b): For Minqin and Dunhuang sites, the AOD440 and rcoarse are shown for "-", the authors may want to indicate the missing data. Please give the explanation in the context. Response: Thank you very much for Reviewer's good comments. We have added "Note that the "-" in Figure 5(b) represents that missing data for AOD440 and rcoarse at Dunhuang and Minqin sites." in Page 34, Lines 912-913.

Minor comments: (1) 1. Introduction, Page 5, line 142: Please add "There have been several world-famous aerosol long-term monitoring networks over Asian region for examining aerosol features and its radiative effects, for instance, AERONET—AErosol RObotic NETwork (Holben et al., 1998), SKYNET—aerosol-cloud-radiation interaction ground-based observation network (Nakajima et al., 1996; Takamura et al., 2004; Che et al., 2008), and CARSNET—China Aerosol Remote Sensing Network (Che et al., 2009, 2014, 2015)." at the beginning of this section and add corresponding cited literature in References. Response: We have added "There have been several world-famous aerosol long-term monitoring networks over Asian region for examining aerosol features and its radiative effects, for instance, AERONET—AErosol RObotic NETwork (Holben et al., 1998), SKYNET—aerosol-cloud-radiation interaction ground-based observation network (Nakajima et al., 1996; Takamura et al., 2004; Che et al., 2008), and CARSNET—China Aerosol Remote Sensing Network (Che et al., 2009a, 2014, 2015)." at the beginning of this section and add corresponding cited literature in

References.

(2) Page 7, line 200: "3. Asian Dust Optical properties" ⇒ Change to "3. Asian Dust Optical Properties" Response: We have changed to "3. Asian Dust Optical Properties" in Page 7, Line 200.

(3) Page 7, lines 201-205: Change to "A great amount of publications have verified that mineral dust aerosols are commonly predominant by large particles with coarse mode (radii>0.6 ïA■m), which are the essential feature differentiating the dust from fine-mode dominated biomass burning and urban-industrial aerosols (Dubovik et al., 2002b; Eck et al., 2005; Bi et al., 2011, 2014; Kim et al., 2011; Che et al., 2013)." Response: We have changed in Page 7, Lines 201-205.

(4) Page 8, line 209: "compared to" ⇒ Change to "compared with" Response: We have changed "compared to" to "compared with" in Line 209.

(5) Page 8, line 219: change "literatures" to "literature" and modify the other places in the whole manuscript. Response: We have changed "literatures" to "literature" in Line 219 and modify the other places in the whole manuscript.

(6) Page 9, lines 259-260: "Note that the occurred months of PDU days are nearly different from TDU days at Dalanzadgad," ⇒ Change to "Note that the occurred months of PDU cases are nearly different from TDU cases at Dalanzadgad," Response: We have changed to "Note that the occurred months of PDU cases are nearly different from TDU cases at Dalanzadgad," in Page 9, Lines 259-260.

(7) Page 10, line 294: "and estimated SSA at 325 nm ( 0.80) is much lower than at 660 nm ( 0.95)." ⇒ Change to "and estimated SSA at 325 nm ( 0.80) is much lower than that at 660 nm ( 0.95)." Response: We have changed "and estimated SSA at 325 nm ( 0.80) is much lower than that at 660 nm ( 0.95)." in Page 10, Line 294.

(8) 5. Summary, Page 17, line 494: change "PUD" to "PDU" and modify the other places in the whole manuscript. Response: We have changed "PUD" to "PDU" in Page

17, Line 494and modify the other places in the whole manuscript.

Please also note the supplement to this comment:
http://www.atmos-chem-phys-discuss.net/acp-2016-764/acp-2016-764-AC2-
supplement.pdf

---

## Author Comment (AC3) · 18 Oct 2016

Manuscript No.: acp-2016-764 Journal: ACP The revised manuscript entitled "Comparison of key absorption and optical properties between pure and transported anthropogenic dust over East and Central Asia." by Jianrong Bi, et al.

Anonymous Referee3: The work presented in this paper is very interesting and well structured. The authors suggest a method for discriminating the presence of Desert dust in the atmosphere, dividing it into two different cases: pure dust and transported anthropogenic dust. The method is based on a threshold on AOD (440 nm) and Angstrom exponent (calculated using the two wavelengths 400 and 870 nm), and provided good results when compared with the plots of the volume size distributions. Also

the section devoted to the comparison among the values retrieved from measurements and the ones from models generally used, is very interesting and useful.

Response to Referee3: We are grateful to the Editor and the anonymous Referee for their constructive and insightful comments. The comments of the Referees are helpful and valuable for greatly improving the manuscript. Please find a point-by-point reply to the issues as follows (highlighted in blue color font). And we have also uploaded the file of "Response to-Referee3(acp-2016-764)-supplement.pdf".

(1) To complete the paper, I suggest the authors to give a look to the following paper where a similar work has been done for Saharan dust in Europe: "Inventory of African desert dust events over the southwestern Iberian Peninsula in 2000-2005 with an AERONET Cimel Sun photometer", Toledano et al., 2007, DOI:10.1029/2006JD008307. Also in this paper thresholds on Angstrom exponent and AOD are used in order to set up an Automatic Criterion for Detection and Evaluation of Desert Dust Intrusions and, as expected, they are different from the ones used in this paper. I think it should be highlighted in the text that the chosen values are good for the type of dust intrusion of the selected area, and that for a smaller area or a different geographical location, they must be selected carefully. In that paper it is also written that a larger sensitivity to the presence of dust particles has been found at 870 nm rather than 440 nm. Do the authors think that using a threshold on this wavelength in the case of TDU would help to discriminate more accurately the amount of dust from the anthropogenic aerosols? Did the author never found (in TDU dataset) a 3 modal volume size distribution? If yes, it could be another possibility for better understanding the composition of TDU dust. Response: Thank you very much for Reviewer's insightful comments. We have read carefully the paper of Toledano et al. [JGR, 2007]. In their Automatic Criterion for Detection and Evaluation of Desert Dust Intrusions, they mainly based on AOD and Angstrom exponent (AOD(870nm)>0.11 and alpha<0.99), manual inspection, and volume concentrations (fine and coarse modes), and confirmation with back trajectories and satellite-constitute the basic methodology to establish the inven-

tory of African dust events. As pointed out by Toledano et al., "In principle the criterion derived here for the detection of desert aerosol events is only valid for our site, that is, it is a local criterion." (Page 11). If we used the threshold on 870 nm wavelength (AOD(870nm)>0.11 and alpha<0.99) to identify the TDU in our manuscript, we found that there were a lot of cases meet this condition, that is, both dust aerosols and the other aerosol types (e.g., urban-industrial aerosol) have got AOD(870nm)>0.11 and alpha<0.99. Therefore, we could not discriminate among the pure dust, TDU, and fine-mode dominated non-dust aerosols in our study. Meanwhile, we have checked the TDU datasets at all selected sites in our paper, we only found that there were only a few TDU cases (less than 1

Minor comments: (1) Line 189: put the acronyms of SSA, ASY, Ri and Re in line 186, where these quantities are listed. Response: Thank you very much for Referee's good comments! We have presented the full name of SSA, ASY, Ri and Re in Lines 145-146, so we used the acronyms here.

(2) Line 192: "are dependent on AOD440>=0.4" I think it would be better saying "are valid for AOD440..." Response: We have changed "are dependent on" to "are valid for" in Line 192.

(3) Line 277: "capability" instead of "intensity" Response: We have changed "capability" to "intensity" in Line 277.

(4) Line 346: it is written that the pick radius of the coarse mode is about 2.24 um for both PDU and TDU. However for Yulin in TDU it seems to be about 3 um. I think that in the case of TDU it would be better saying that the pick radius in between 2-3 um. Response: Thank you very much for Referee's insightful comments! The rVc of TDU cases actually vary between 2 to 3 um. So, we have changed "for all PDU and TDU cases" to "for all PDU cases and rVc 2.0-3.0 um for TDU cases" in Line 346.

(5) Lines 512-514: the sentence begins with "because" but it doesn't seem to have a correct grammatical structure (subject, verb, object...). Please check it. Response:

[Figure]

We have deleted "because" and changed to "It is very difficult to quantify the anthropogenic contribution due to large uncertainties in defining. . ." in Lines 512-514.

Please also note the supplement to this comment:
http://www.atmos-chem-phys-discuss.net/acp-2016-764/acp-2016-764-AC3-supplement.pdf

————————————————————